# Hygroscopicity of Gel-Forming Composite Materials: Thermodynamic Assessment and Technological Significance

Andrey V. Smagin [1,2,3,*], Nadezhda B. Sadovnikova [1] and Elena A. Belyaeva [2]

1    Soil Science Department, Lomonosov Moscow State University, GSP-1, Leninskie Gory,
     119991 Moscow, Russia
2    Institute of Forest Science, Russian Academy of Sciences (ILAN), 21, Sovetskaya, 143030 Moscow, Russia
3    Center for Mathematical Modeling and Design of Sustainable Ecosystems, Peoples' Friendship University of
     Russia Miklukho-Maklaya 6, 117198 Moscow, Russia
*    Correspondence: smagin@list.ru; Tel.: +7-(495)-916-913-79-48

**Abstract:** Hygroscopicity is an important technological property of composite materials for the conservation and treatment of water in modern technologies for sustainable green environment and agriculture. Using a thermodynamic approach, this study analyzes the hygroscopicity of composite gel-forming soil conditioners as a function of water activity and temperature. A simple and generally available method of water thermo-desorption is proposed for the quantitative assessment of hygroscopicity, dispersity and potential resistance of composite materials to osmotic collapse. It is based on the fundamental thermodynamic dependence of water potential and temperature of the dried material in a thermodynamic reservoir (laboratory) with constant relative humidity. The hygroscopicity of the studied composite materials in humid air (relative humidity over 90%) reaches a water content of 80–130% (wt); however, this water has too high retention energy and cannot be consumed by green plants, which calls into question the technology of obtaining water from the air using hygroscopic materials. The high hygroscopicity of hydrogels and its dynamics, depending on the controlling factors of temperature and air humidity, must necessarily be taken into account in the materials trade and in the technological calculation of doses for the use of these materials in sustainable agriculture and landscaping.

**Keywords:** composite materials; water thermodynamics; hygroscopicity; dispersity; water retention energy; osmotic stress; stability of gel structures

## 1. Introduction

More than 70% of our "blue planet" is covered by water. However, this sea water with a high (34–36‰) salt concentration is absolutely unsuitable for the vast majority of higher plants and animals, as well as for technological processes that ensure the stability of our civilization. Sustainable development requires about 500 m$^3$ of water per person annually; thus, the shortage of freshwater resources is one of the most acute environmental problems on the planet, perhaps more serious than the depletion of oil, coal and gas reserves. The greatest consumption of clean water is in agriculture, especially in arid climatic conditions with high daily evapotranspiration (12–15 L/m$^2$) and unproductive water infiltration losses exceeding 50% of the irrigation norm in widespread arenosols [1]. Innovative composite materials can be used both in water purification technologies and for water conservation in the soil by reducing unproductive losses due to evaporation and infiltration [2–11]. Relatively small doses (0.1–0.6% (wt)) of gel-forming soil conditioners based on weakly cross-linked polyacrylamide, copolymers of acrylamide and acrylic acid salts, their compositions with mineral fillers (phyllosilicates, zeolites), amphiphilic biopolymers from natural raw materials (humates, lignin, peat), intercalated with ions of monovalent metals (sodium, potassium, lithium), increase the water retention of sandy, loamy-sandy and sandy-loamy arenosols by 3–5 times or more, increase the reserves of productive water in topsoil by

1.5–2 times or more, and reduce water evaporation by 1.3–3 times and unproductive in-filtration losses by 3–10 times [2,4–7]. The use of gel-forming superabsorbents provides a 1.5–2-fold or more increase in seed germination, prolongation of the survival time of grass and woody plants under water stress, growth of aboveground and underground biomass of plants and their total yield, along with the possibility of 1.3–2-fold saving of water resources [5–7]. Despite the promising positioning of natural gel-forming biopolymers, mainly polysaccharides, synthetic superabsorbent polymers displace natural ones due to their high absorbency, availability of a wide range of raw materials and longer service life [8]. For finely dispersed soils, a small additive (10 mg/L) of water-soluble anionic polyacrylamide in irrigation water significantly affects infiltration and water absorption from irrigation furrows, structuring and protecting the soil from water erosion [3].

Along with the optimization of the water regime, gel-forming composite materials can be successfully used as agents for systems of retention and controlled release of agrochemicals and pesticides [7,8,10]. A sharp (up to 10 times or more) reduction of infiltration water losses, along with the retention of water-soluble fertilizers (NPK), trace elements and pesticides in the liquid phase inside the polymer mesh, as well as ion exchange and adsorption due to chemical and physical interaction with the polymer matrix and its fillers, reliably protect the active substance from leaching [7,8]. Intercalation of biocides (silver ions and nanoparticles, organic fungicide) in composite acrylic superabsorbents, as well as biomimetic technologies for the synthesis of polymethacrylates with a cationic and membranolytic biocidal effect, allow for obtaining materials for environmentally friendly control of pathogens and eutrophits with low effective concentrations $E_{50}$ of biocides near 10–100 ppm (soil) and 100–6000 ppb (water) [7,10]. Technologies of nanostructural organization and architecture of composite polymer materials, as well as the synthesis of smart gels, are actively used to create various forms of composite materials (microgels, nanocapsules, films, nanosheets, multilayer gel membranes, etc.), including polymers with a changing structure for controlled release of the retained substance (nutrients, water) under the influence of temperature, pH, electricity and other controlling factors [8,9,11,12]. For example, MXene nanosheets with electrochemical regeneration can be used for adsorptive removal of organic antibiotics such as ciprofloxacin from wastewater [9], and poly(N-isopropylacrylamide)-(PNIPAm-)-based thermoresponsive microgels and hydrogels are proposed to be used both for the removal of pollutants in wastewater treatment and for seawater desalination [11]. Another new direction explores the possibility of water vapor absorption from the air by composite materials for its concentration and accumulation in liquid form [11,12]. In any case, the effectiveness of innovations should be checked by a strict thermodynamic calculation based on a thermodynamic assessment of the water potential or the work that must be expended to extract water from a composite material [13].

This study uses a thermodynamic approach to assess the hygroscopicity of gel-forming composite materials for water retention in soils and water conservation in arid irrigated agriculture and green landscaping. The research is aimed at obtaining experimental dependencies of the hygroscopicity and activity (potential) of water in these materials in a wide range of temperature and relative air humidity inherent in arid climates. For this purpose, we propose a simple and, generally accessible for any laboratory, thermodesorption method based on the fundamental thermodynamic dependence of the activity (potential) of water on the drying temperature of the material obtained from the Clausius–Clapeyron equation [14]. Approximation of the experimental data by the fundamental model of the Deryagin disjoining water pressure [13] makes it possible to simultaneously estimate the dispersity (specific surface area) and potential resistance of composite gel-forming materials to osmotic coagulation (collapse) of the gel structure. The most important result of the study is experimental information concerning the high values of maximum hygroscopicity (80–130 wt.% and more) of gel-forming soil conditioners, as well as the wide range of possible variation of this indicator in climatic conditions with contrasting temperature and humidity. This result must necessarily be taken into account in calculating the correct doses for the practical application of the studied materials and in their trade. The discussion

part of the publication uses the obtained results and the basic thermodynamic approach to critically analyze the effectiveness of innovative technologies for obtaining water from air using hygroscopic composite materials.

## 2. Materials and Methods

### 2.1. Tested Composite Gel-Forming Materials

Innovative composite materials were synthesized at the Ural Chemical factory (Perm, Russian Federation) under the "Aquapastus" trademark using our patented technology [7] (see also "Patents" section at the end of the article). A mandatory hydrophilic base is an acrylic polymer matrix with a varying ratio of copolymers of acrylamide and acrylic acid salts (ammonium and sodium acrylates). In the base hydrophilic material Aquapastus-11 (further abbreviated A11), an acrylic polymer matrix is filled by 23% (wt) biocatalytic wastes from the production of polyacrylamide. In another innovative material Aquapastus-22-Ag (A22-Ag or «black» gel), a similar polymer matrix was filled by 23% (wt) dispersed peat and by 1% silver ions, for the material's resistance to biodegradation and osmotic stress. This hydrogel also could presumably form more stable to pressure gel structures of the reinforced type due to the fine-dispersed peat filler. The innovative composite materials were compared with each other, as well as with well-known brands of composite acrylic materials "Aquasorb" (SNF-group, [14]); "Zeba" (UPL-group, [15]) based on polyacrylamide, acrylic acid and starch.

### 2.2. Thermo-Desorption Instrumental Method for Laboratory Testing of Composite Gel-Forming Materials

Our new method uses the thermodynamic dependence of the potential (activity) of water on the drying temperature of the material, obtained from the fundamental Clausius–Clapeyron equation in [16]:

$$|\psi| = Q - \beta \cdot T, \tag{1}$$

where $|\psi|$, (J/kg) is the absolute value of water potential, $\beta = \{Q/T_r - R \cdot \ln(f_r)/M\}$, $Q = 2401 \pm 3$ kJ/kg is the specific heat of evaporation for the temperature range of 0–105 °C, $R = 8.314$ J/(mol·K) is the universal gas constant, $T$, (K) is an absolute temperature in the drying oven, $M = 0.018$ kg/mol is the molar mass of water. This dependence assumes that under the conditions of a thermodynamic reservoir (laboratory) with constant relative air humidity ($f_r$) and room temperature ($T_r$), local heating will increase the saturation vapor pressure ($P_0$) in relation to the total vapor pressure ($P$) and, accordingly, decrease water activity ($f = P/P_0$) in the drying oven. The hygroscopicity or water content in the material ($W$, (%)) will also decrease upon heating, therefore, using the drying of the material at different stages from room temperature ($T_r = 293$ K) to the standard temperature of conditionally complete dehydration of the material (105 °C) and weighing the sample at each stage—step by step, it is possible to estimate dependence $\psi(W)$ by a sequence of experimental points. For stepwise drying of samples, any temperature-controlled drying oven, or in the express version of the experiment, a laboratory microwave moisture analyzer can be used. This study used a BINDER ED023-230V drying oven (Germany, BINDER GmbH, Tuttlingen) and an AND MX-50 humidity analyzer (Japan, A&D, Tokyo). Samples of composite materials weighing 3–5 g in glass vials were exposed in a drying oven at a given temperature until thermodynamic equilibrium (2–3 h) and then quickly (to avoid reverse adsorption of vapor from laboratory air) were weighed on an accurate (0.0001 g) laboratory scales OHAUS Pioneer PX224/E (Switzerland, OHAUS, Greifensee). Calculation of hygroscopicity (water content $W$, (%)) was carried out according to the formula:

$$W = 100 \frac{m_t - m_s}{m_s - m_0}, \tag{2}$$

where $m_t$, (g) is the mass of a vial with hygroscopic material at a given drying temperature, $m_s$, (g) is a similar value at a temperature of 105 °C for conditionally complete dehydration

of the material, $m_0$, (g) is the mass of an empty vial. All experiments were carried out in triplicate.

### 2.3. Additional Methods and Data Processing

To additionally control the dynamics of temperature and relative humidity in the drying oven, programmable data loggers DS1923 (USA, Maxim Integrated Products, Inc. Wilmington, MA) were used. Standard gravimetric analysis of water absorption (equilibrium degree of swelling) of gel-forming composite materials was carried out in distilled water and sodium chloride solutions with a concentration of 0.1; 0.5; 1; 2; 3 g/L. Statistical and mathematical processing of the results, including data approximation by nonlinear thermodynamic models, was carried out using MS Excel, Microsoft Office 2016 (USA, Microsoft, Redmond, DC) and S-Plot 11 (Germany, Systat Software GmbH, Erkrath) computer software.

## 3. Results

### 3.1. Theoretical and Methodological Basis for the Thermodynamic Assessment of Hygroscopicity in Polydisperse Materials

Hygroscopicity is conveniently estimated by the equilibrium water content in the material ($W$). This indicator is dynamic, and it is controlled by the dispersity and surface energy of the material and by the thermodynamic conditions of air temperature and humidity. All these factors are taken into account by the dependence of the water content in the material and the thermodynamic potential (activity) of water ($\psi(W)$ or $W(f)$). The relationship between water activity ($f$) and thermodynamic potential at constant temperature ($T$) is given by the well-known Gibbs formula [13]:

$$\Psi = \frac{RT}{M} \ln(f), \tag{3}$$

(for notation used, see Formula (1)).

For a thermodynamic reservoir with a constant relative humidity of the air, and, accordingly, the activity of water ($f = f_r$), but a variable temperature of the dried material, the thermodynamic potential of water is estimated by the fundamental formula (1). Its verification using DS1923 air temperature and humidity loggers shows complete identity with the experimental data (Figure 1) and confirms the physical validity of the new method of thermo-desorption assessment of hygroscopicity for drying materials. According to Polanyi's fundamental theory, the thermodynamic dependence $\psi(W)$, or the so-called Polanyi potential curve, is temperature invariant [16]. This means that the water vapor desorption isotherms $W(f)$ can be calculated from the $\psi(W)$ curves for any temperature using the formula inverse to (3):

$$f = exp\left(\frac{\Psi M}{RT}\right), \tag{4}$$

Isotherms $W(f)$ at room temperature ($T = 293$ K) can be approximated by the standard BET model [17]:

$$W = \frac{W_m K \cdot f}{(1 - f) \cdot (1 + (K - 1) \cdot f)}, \tag{5}$$

where $W_m$ is the water content at the monomolecular coating of the material surface (monolayer BET capacity), $K$ is the BET polymolecular sorption constant. After that, it is easy to evaluate the dispersity of the material or the specific surface area ($S_{BET}$, (m$^2$/g)) [15,16].

$$S_{BET} = \frac{W_m N_A \cdot s_0}{M} \approx 3515 \cdot W_m, \tag{6}$$

where $N_A$ (mol$^{-1}$) is Avogadro's number, $s_0$ (m$^2$) is the area covered by one water molecule.

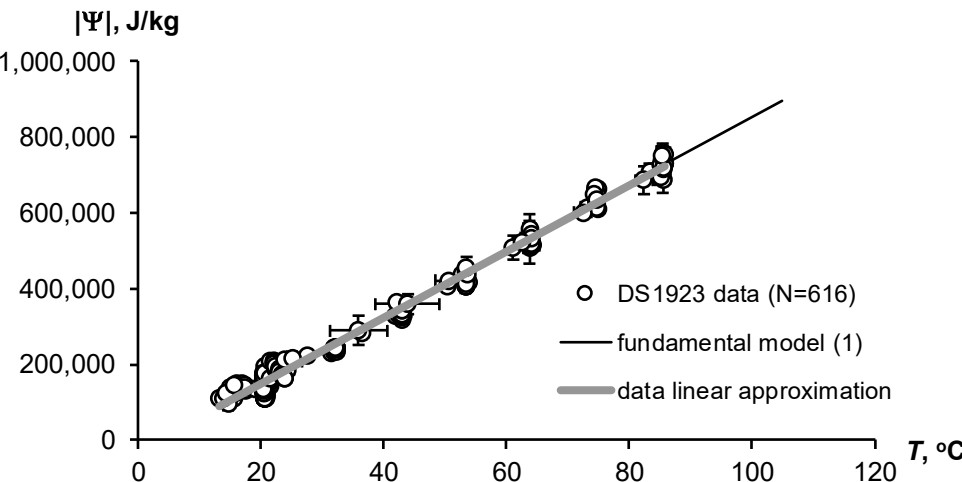

**Figure 1.** Experimental dependence of the water potential and temperature in the drying oven (estimated by data loggers DS 923) and its modeling.

Our alternative approach for assessing the dispersity of materials uses the fundamental model of the Deryagin disjoining pressure to describe the thermodynamic dependence $\psi(W)$, according to [13,18]:

$$|\Psi| = a \cdot exp\left(-\frac{h}{\lambda}\right) = a \cdot exp(-b \cdot W), \quad b = \frac{1}{S\rho\lambda}, \tag{7}$$

where $a$ (J/kg) is the physically based parameter reflecting the surface shape and potential (charge), $h$ (m) is the thickness of the water film on the surface of the material, $\lambda$ (m) is the length of correlation for the structural forces or effective Debye thickness of the double electric layer for ion-electrostatic forces, $\rho$ (kg/m$^3$) is the density of water, $S$ (m$^2$/kg) is the variable-specific surface of the interphase boundary, $b$ (kg/kg) is the physically based parameter controlled by $\lambda$ and $S$ according to the modern concept of thermodynamics of water retention and dispersity in soils [18].

In a standard stable state with a minimum water film thickness ($h_{st} = 2\lambda_{st}$), the specific surface area ($S_{st}$) is defined by the slope of the water retention curve $\psi(W)$ as [13]:

$$S_{st} = \frac{1}{br_0\rho}\sqrt{2exp(-2)} \approx \frac{1}{2br_0\rho}, \tag{8}$$

where $r_0 = 1.38 \times 10^{-10}$ m is a crystallochemical radius of water molecule. The new approach gives estimates of $S_{st} \approx S_{BET}$ close to the standard BET assessment in a simpler way, regardless of temperature, according to the temperature invariance of Polanyi's thermodynamic curve $\psi(W)$.

The fundamental disjoining pressure model (8) is also used to estimate the generalized Hamaker constant ($A_G$, (J)) for the molecular (dispersive) component of the disjoining pressure [13,16]:

$$A_G = \frac{24a\pi r_0\rho}{\sqrt{2exp(-2)}} \approx 48a\pi r_0\rho; \tag{9}$$

critical water content of the particle coagulation threshold in two-phased gel systems [13]:

$$W_{cr} = 2/b; \tag{10}$$

and the critical concentration corresponding to the coagulation threshold ($C_{cr}$, (mol/L)) or gel structure collapse by electrolytes in the intermicellar solution [19]:

$$C_{cr} \approx 213 \frac{(RT)^5 (\xi_0 \xi)^3}{A_G^2 (Fz)^6}, \qquad (11)$$

where $F$ (C/mol) is the Faraday number, $z$ is the ion valence, $\xi_0$ (F/m) is the electric constant, $\xi$ (dimensionless) is the dielectric permittivity of the dispersion medium (water). The product $W_{cr} \cdot C_{cr}$, (mmol/kg) represents the critical concentration of the electrolyte, causing the total collapse of the gel structure, relative to the dry weight of the polymer composite material or soil substrate.

### 3.2. Experimental Results and Their Modeling

Figure 2A represents the experimental thermodynamic dependences of the water potential and the water content in the studied composite materials. The water vapor desorption isotherms calculated from these data at a standard room temperature of 293 K are shown in Figure 2B. Both figures reveal a high maximum hygroscopicity of gel-forming composite materials, reaching 80–130% water content if the water activity tends to unify. The innovative composite material A22-Ag with an amphiphilic filler of an acrylic polymer matrix in the form of dispersed peat and the addition of ionic silver has the maximum hygroscopicity values. The lowest hygroscopicity values are observed in the composite soil conditioner "Zeba" based on polyacrylamide and starch. Other materials are intermediate in terms of water retention, which is reflected by the corresponding curves $\psi(W)$ and $W(f)$ in Figure 2A,B.

The fundamental model (7) adequately describes the experimental data $\psi(W)$ and $W(f)$, which is confirmed visually (dotted lines in Figure 2A,B) and analytically from the statistical reports of the nonlinear approximation given in Table 1.

**Table 1.** Approximation indicators for fundamental models (7) and (5).

| Materials: | Approximation Parameters | | | Statistical Parameters | |
|---|---|---|---|---|---|
| | Model (7): $|\psi| = a \times \exp(-bW)$ | | | | |
| | $a$, (J/kg) | $b$, (kg/kg) | $R^2$ | $s$, (J/kg) | $p$ Value * |
| "Zeba" | $998457 \pm 12520$ | $0.168 \pm 0.004$ | 0.998 | 15405 | <0.0001 |
| "Aquasorb" | $822805 \pm 16769$ | $0.060 \pm 0.002$ | 0.998 | 21859 | <0.0001 |
| A11 | $789655 \pm 16308$ | $0.063 \pm 0.002$ | 0.998 | 19673 | <0.0001 |
| A22-Ag | $742452 \pm 16768$ | $0.043 \pm 0.003$ | 0.996 | 21857 | <0.0001 |
| | Model (5): $W = W_m \times K \times f / \{(1 - f) \times (1 + (K - 1) \times f\}$ | | | | |
| | $W_m$, (%) | $K$ | $R^2$ | $s$, (%) | $p$ value * |
| "Zeba" | $7.42 \pm 0.26$ | $54.32 \pm 12.11$ | 0.983 | 0.68 | 0.0028 |
| "Aquasorb" | $18.71 \pm 0.96$ | $30.14 \pm 9.02$ | 0.971 | 2.32 | 0.0124 |
| A11 | $16.86 \pm 0.62$ | $29.80 \pm 6.32$ | 0.985 | 1.48 | 0.0022 |
| A22-Ag | $23.83 \pm 0.50$ | $24.69 \pm 2.89$ | 0.996 | 1.15 | 0.0010 |

* the worst $p$ value (maximum) for one of the two model parameters; $\pm$ indicates the confidence interval of the model parameter for a given $p$ value.

High coefficients of determination $R^2 = 0.996$–0.998 and small standard errors s = 15–20 kJ/kg, as well as the statistical significance of both physically based parameters $a$, $b$ at $p$ value $\leq 0.001$ convince of the complete adequacy of the model (7). For comparison, the standard BET model is less suitable even for the usual BET range of water activity $f \leq 0.4$ ($R^2 = 0.971$–0.996; s = 0.7–2.3%; $p = 0.001$–0.0028); for the entire range of water vapor desorption isotherms $f \leq 0.98$, it is completely unsuitable (solid gray lines in Figure 2B), which is well known [17].

Another advantage of the fundamental model (7) is the possibility of estimating important technological indicators of dispersity, surface energy and stability of gel-forming composite materials in accordance with formulas (8)–(11). The corresponding estimates differ statistically significantly ($p$ value $\leq 0.01$) for the studied composite materials (Figure 3A).

The maximum dispersion estimates (840–880 m$^2$/g) both by the BET model and by the disjoining pressure model (7) are revealed for the innovative material A22-Ag. Composite soil conditioner "Zeba" has a minimum specific surface (230–260 m$^2$/g), as well as a minimum water absorption (hygroscopicity). The European brand "Aquasorb" and the composite material A11 are also intermediate in terms of dispersity characteristics. Estimates of the specific surface area by the standard BET method and by the new model (7) give similar results for both materials, which do not differ statistically significantly taking into account their variation (Figure 3B).

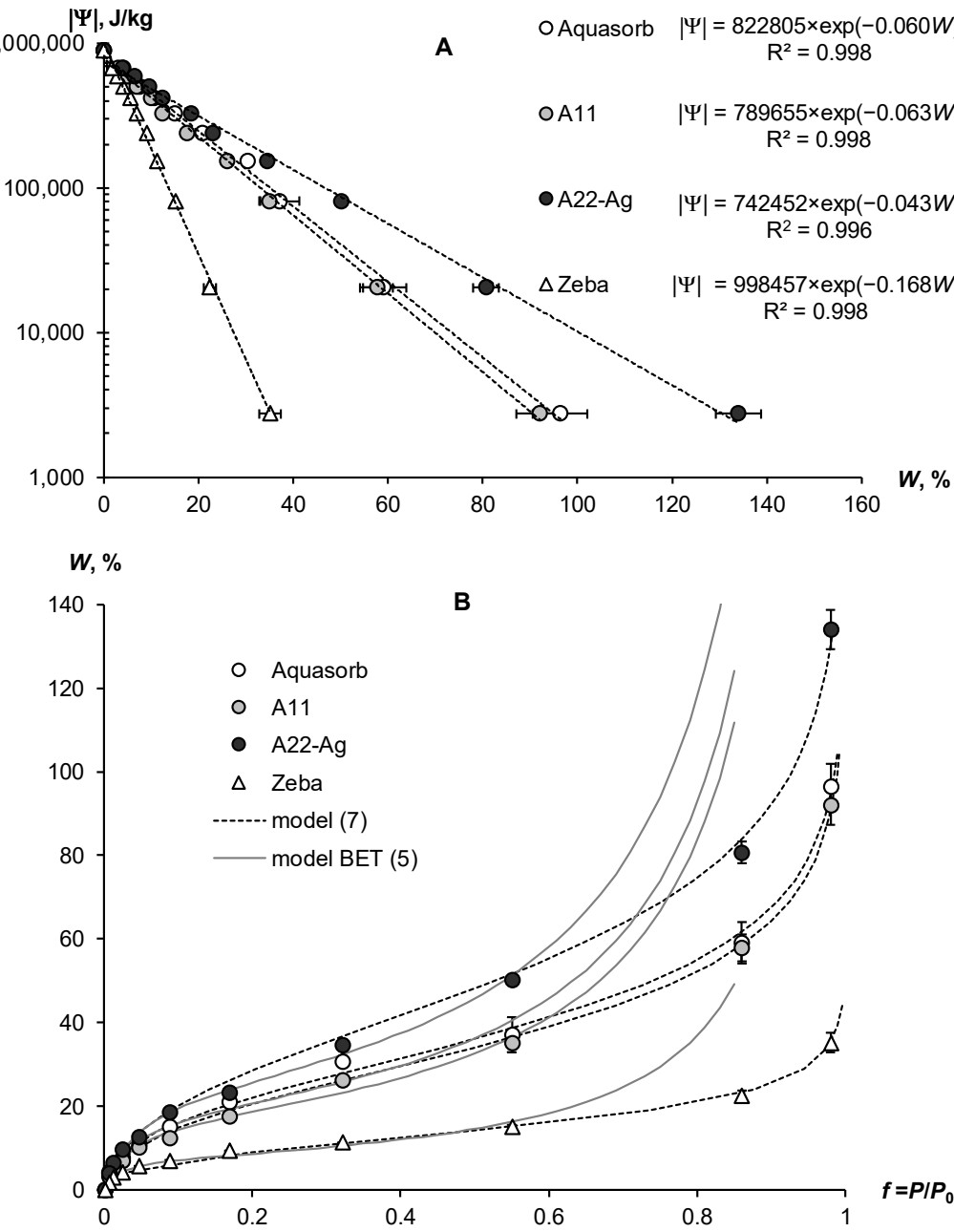

**Figure 2.** Experimental dependences of the thermodynamic potential and water content (**A**) and calculated water vapor desorption isotherms (**B**). Approximations of the experimental data by the disjoining pressure model (7) and the standard BET model (5) are shown by black dotted and solid gray lines, respectively.

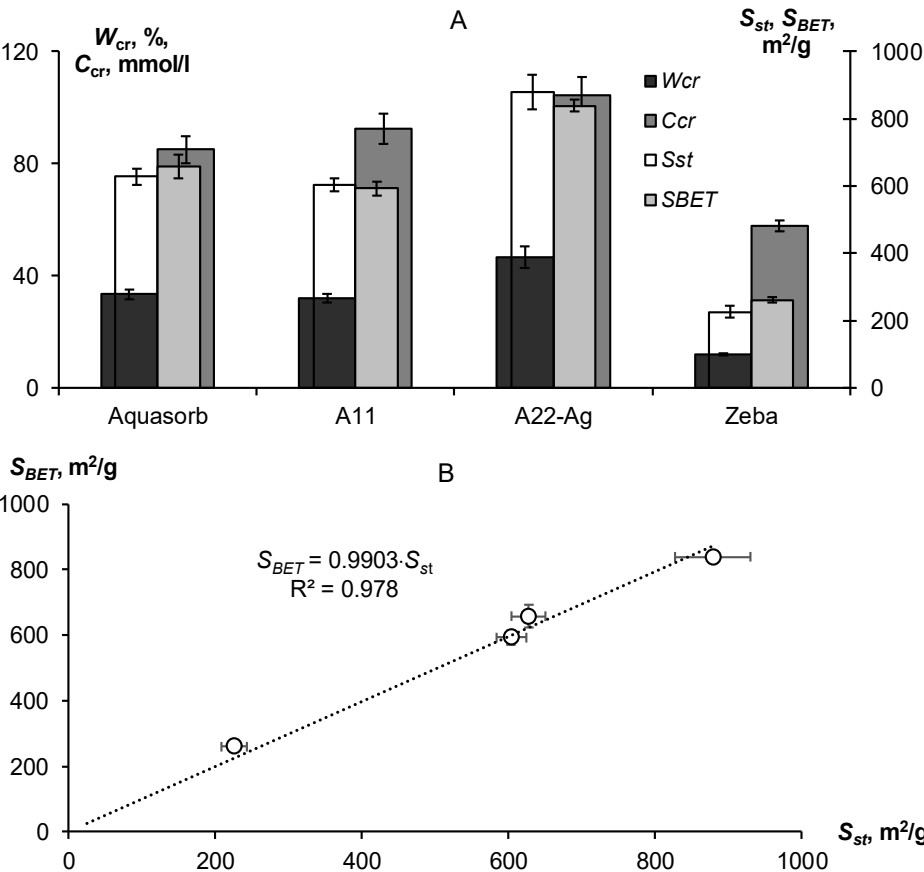

**Figure 3.** Indicators of dispersity of composite materials and stability of gel structures (**A**). Comparison of dispersion estimates for models (6) and (8) (**B**).

The Hamaker constants change rather weakly in a number of compared composite materials: $3.1 \times 10^{-19}$ J ("Aquasorb"); $3.0 \times 10^{-19}$ J (A11); $2.8 \times 10^{-19}$ J (A22-Ag); $3.8 \times 10^{-19}$ J ("Zeba"). The critical concentration of a monovalent ($z = 1$) binary electrolyte for the collapse of gel structures is maximal for the new A22-Ag material ($C_{cr} = 104 \pm 7$ mmol/L) and minimal for the Asian brand "Zeba" ($C_{cr} = 58 \pm 2$ mmol/L) (Figure 3A). A more detailed criterion $C_{cr} \cdot W_{cr}$ also shows the lowest resistance to electrolytes for the "Zeba" material ($7 \pm 1$ mmol/kg), and the maximum resistance for the A22-Ag hydrogel ($49 \pm 3$ mmol/kg), while the "Aquasorb" and the A11 materials are characterized by similar values of this indicator ($28 \pm 2$ and $30 \pm 2$ mmol/kg). The A22-Ag material, according to the theoretical criteria $C_{cr}$ and $W_{cr} \cdot C_{cr}$, should be the most resistant to the osmotic collapse of gel structures, since its composite matrix is able to swell in solutions of monovalent electrolytes with a concentration of up to 100 mmol/L. In terms of widespread in arid climates sodium chloride or "marine" salinization, taking into account the molar mass of NaCl M = 58.5 g/mol, the collapse threshold for this material will be equal to 6 g/L NaCl. For other materials, the same indicator varies within 3.4–5 g/L. This theoretical analysis is confirmed by a direct experimental evaluation of the swelling of hydrogels in sodium chloride solutions of different concentrations (Figure 4). The degree of swelling of the most stable composite material A22-Ag in NaCl solution with a maximum concentration of 3 g/L decreased to $153 \pm 23$ g/g, and in the least stable hydrogel "Zeba" to $15 \pm 8$ g/g. Hydrogel "Aquasorb" and A11 occupied an intermediate position with varying the residual degree of swelling from 53 to 86 g/g.

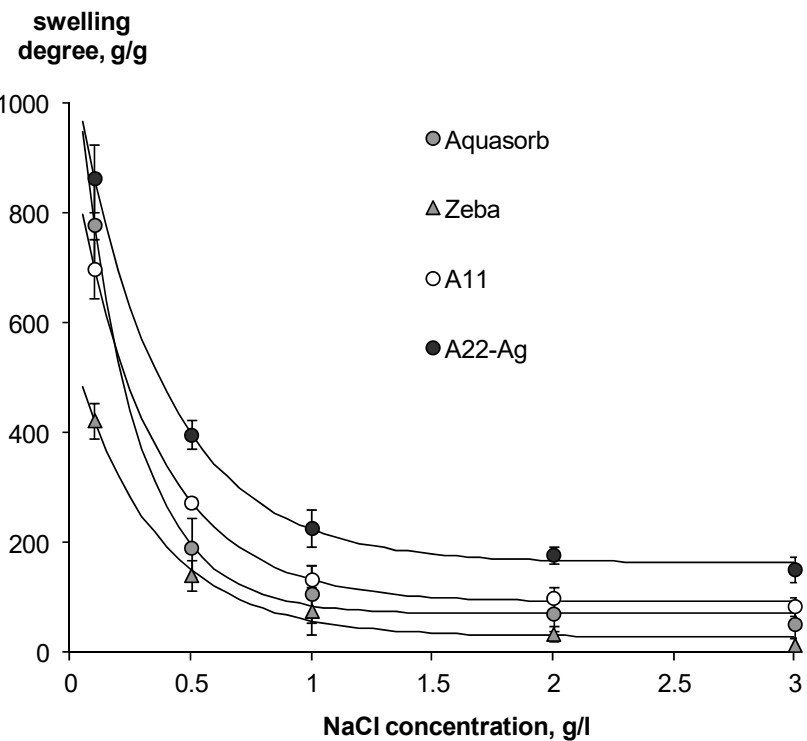

**Figure 4.** Swelling of gel-forming soil conditioners in sodium chloride solutions.

To assess the effect of soil salinity on the stability of gel-forming soil conditioners, the $W_{cr} \cdot C_{cr}$ indicator, which recalculates the critical salt content relative to the mass of the solid phase of the material, is more suitable. Its maximum value of $49 \pm 3$ mmol/kg obtained for the A22-Ag hydrogel, or about 3 g/kg for sodium chloride salinity, corresponds to a moderate soil salinity level of no more than 0.3% [1]. In more saline soils, the investigated composite gel-forming materials cannot be used successfully to increase soil water retention and conserve water resources.

## 4. Discussion and Practical Use of Results

### 4.1. Comparison of the Obtained Results with Known Data

The results of high (80–100% and more) hygroscopicity of the studied composite hydrogels exceed similar indicators for natural colloids, for example, clay soils and minerals (15–40%), wood and caustobiolites (20–60%), organic food, seeds (15–30%), according to [17,20–24]. However, they are inferior in hygroscopicity to the innovative development [12] based on the polysaccharide glucomannan, hydroxypropylcellulose and lithium ions, where the equilibrium water content reaches $W = 60$–150% in the range of water activity $f = 0.3$–0.6. The innovative composite material A22-Ag with an amphiphilic filler of an acrylic polymer matrix in the form of dispersed peat and the addition of ionic silver has the maximum hygroscopicity values more than 100% at $f \to 1$, and in the range $f = 0.3$–0.6, its hygroscopicity does not exceed $W = 35$–60%.

The dispersity estimate with a range of 230–880 m$^2$/g corresponds to the upper limit of the specific surface for fine-dispersed soils—chernozems and montmorillonite vertisols (150–400 m$^2$/g), peat, humus and clay minerals (200–800 m$^2$/g), according to [17,21,25]. The result of the coincidence of estimates of the specific surface of colloidal-dispersed porous materials by the BET method and by the fundamental model (7) was previously obtained in [18].

The increase in hygroscopicity, water retention, dispersity and resistance to salt collapse closely depends on the components introduced into the acrylic polymer matrix of composite materials. The introduction of amphiphilic ionogenic groups with monovalent metal ions (sodium, lithium, silver) that expand the double electric layer contributes to

higher water absorption and resistance of gel structures to osmotic collapse [7,26,27]. Hydrophilic biopolymer additives (starch, cellulose), in contrast to amphiphilic fillers, clearly worsen the water absorption of composite materials, reducing their degree of swelling and hygroscopicity. Polysaccharide-based biopolymer hydrogels are more suitable for controlled release systems rather than water retention in soils where superabsorbents are required [28]. Osmotic collapse of gel structures is recognized as one of the most serious environmental factors that limits the effectiveness of gel-forming soil conditioners [6,7,29–32]. Therefore, future innovations in the field of composite materials for a sustainable green environment and agriculture should be aimed at this challenge, especially important for arid coastal areas.

### 4.2. The Debatable Problem of Obtaining Water from the Air by Hygroscopic Materials

The high hygroscopicity of composite gel-forming materials theoretically means the possibility of absorbing large amounts of water from air, especially saturated with water vapor ($f \to 1$). However, from our point of view, different from [12], such technologies are unlikely to be effective in practice. Their failure follows from the analysis of the water potential, according to the fundamental formula of Gibbs (3). Even close to the air saturation state $f = 0.98$ (the standard in soil science for estimating maximum hygroscopicity by adsorption of water vapor from the atmosphere over a saturated solution of potassium sulfate) gives the absolute value of the water potential $|\psi| = 2734$ J/kg at 293 K. This adsorbed water will be even less available for consumption by terrestrial living organisms than sea water with an osmotic pressure of 24–26 bar or an equivalent potential $|\psi| = 2400$–2600 J/kg. The root potential of most agricultural crops does not exceed 2000 J/kg, and the Richards–Weaver estimate of the so-called wilting point, which is standard for soil science, operates with an even lower absolute water potential $|\psi| = 1500$ J/kg [33]. Under more realistic for arid climate conditions of unsaturated air with water activity $f = 0.2$–0.5, the thermodynamic potential of adsorbed water will reach 100,000–200,000 J/kg, i.e., to extract it from a composite material, it is necessary to spend more than 100 kJ per liter of water. In [12], for this purpose, it is supposed to heat the composite material up to 60 °C. According to the fundamental equation (1), confirmed by experimental results (Figure 1), this heating in an atmosphere with water activity $f_r = 0.2$–0.5 will be equivalent to an absolute water potential of 434–575 kJ/kg, which is at least twice the potential of adsorbed water 100–200 kJ/kg; hence, this water will be easily removed from the material. However, the energy costs for heat input of 400–600 kJ for each liter of extracted water and for its subsequent cooling for condensation, according to the technological scheme [12], will be 150–300 times higher compared to the widespread distillation of sea water with an absolute potential of 2.4–2.6 kJ per liter (kilogram) of water. In addition, the rate of absorption of water from the air, limited by the diffusion of water vapor in the adsorbent, is extremely small and does not allow for one to obtain any significant amounts of water from the air without its forced supply to the material. The usual concentration of water vapor in the atmosphere rarely exceeds 20 g/m$^3$; therefore, in order to adsorb 1 L of water, at least 50 m$^3$ of air must be forced into the hygroscopic material, or it is necessary to wait a long time until this material dries 50 m$^3$ of air through the diffusion of water vapor.

### 4.3. Practical Application of the Results

Concluding the discussion of the results, it is necessary to pay attention to their more practical use, in our opinion, than obtaining water from air. This is the exact dosage of gel-forming soil conditioners to improve water retention and conserve water resources in sustainable irrigated agriculture and green landscaping. Previous research indicates that cost-effective soil conditioning requires small doses of composite gel forming materials from 0.05 to 0.3% (wt) [2–7,27]. What do these doses mean? A priori, the dose of a solid soil conditioner ($D$, (%)) is the mass of its solid phase ($m_{sc}$) relative to the mass of dry soil ($m_s$): $D = 100\, m_{sc}/m_s$. However, in practice, when calculating the working dose of ameliorants, their hygroscopicity is often neglected. Analysis of experimental data (Figure 2A,B) shows

that this simplified method is not valid for highly hygroscopic gel-forming materials. The hydrogel powder extracted from the sealed package adsorbs or desorbs water, reaching thermodynamic equilibrium with water vapor in the atmosphere. During the year in a temperate climate, the relative humidity of indoor air usually varies from $f_r$ = 0.2–0.3 (heating winter season) to $f_r$ = 0.6–0.8 and higher (summer, rainy periods). Even more contrasting changes in humidity and air temperature can occur in an arid climate, especially in coastal areas (for example, the Gulf countries), where the amplitude of daily temperatures reaches 30–40 °C, and a change in wind direction leads to a variation in air humidity from $f$ = 0.1–0.2 (wind from the desert) to $f$ > 0.9 (wind from the sea). Indoor air conditioning smooths out these contrasts, but fluctuations in water activity still remain high. Using the well-known formula for calculating the mass of the solid phase of the dehydrated material ($m_{sc}$) from its water content (hygroscopicity $W$, (%)) and the mass in the hygroscopic state with adsorbed water ($m_w$):

$$m_{sc} = m_w \frac{100}{100 + W\%}, \tag{12}$$

it is easy to estimate the error in determining the dose of the material without taking into account its hygroscopicity:

$$E_r = \frac{m_w}{m_{sc}} = \frac{100 + W\%}{100}, \tag{13}$$

where $E_r$ is the factor of underestimation of the real dose of the material due to its dilution with adsorbed water. Thus, for the maximum estimates of the hygroscopicity of the studied gel-forming composite materials $W$ = 80–130% in a humid atmosphere, the $E_r$ index reaches 1.8–2.3 times, which means a similar high underestimation of the actual dose of soil conditioner. In a relatively dry atmosphere ($f \leq 0.5$) with an equilibrium hygroscopicity $W \leq$ 20–40%, according to the data (Figure 3), the underestimation of the dose $E_r$ = 1.2–1.4, which is also quite significant.

By analogy, if the conditions for drying, packaging and commercial hygroscopy of gel-forming soil conditioners are not standardized, the buyer of this product in conditions of high air humidity risks spending half the money not on the composite polymer material, but on the adsorbed water in it.

## 5. Conclusions

The thermodynamic approach and a new thermo-desorption method for studying the water-sorption properties of gel-forming composite materials reveal their high hygroscopicity and the dependence of the mass of materials on the controlling factors of temperature and humidity of the ambient air. The analysis of this dependence on the basis of the fundamental model of disjoining water pressure allowed us to simultaneously assess the dispersity (specific surface area), energy indicators of interfacial interactions and predict the resistance of gel structures to osmotic collapse. For the most promising gel-forming soil conditioners, high hygroscopicity (>100% (wt)) and dispersity (>800 m$^2$g) determine the high efficiency of water retention and the possibility of preserving soil water due to the formation of stable gel structures not only in pure water, but also in saline solutions (TDS = 1–3 g/L). High hygroscopicity does not mean that water can be easily obtained from the air, since the extraction of 1 L of adsorbed water from highly hygroscopic composite materials exceeds 100 kJ. The correct calculation of the working doses of these materials, as well as their weight in commercial transactions, requires mandatory consideration of their hygroscopicity and standardization of the permissible amount of adsorbed water in commercial products.

## 6. Patents

The results of the work are used in the synthesis technology of biodegradation-resistant filled hydrogels patented in the Russian Federation:

Patent RU no. 2726561 (https://findpatent.ru/patent/272/2726561.html (accessed on 22 July 2022)).

Patent RU no. 2639789 (http://www.findpatent.ru/patent/263/2639789.html (accessed on 22 July 2022)).

**Author Contributions:** Conceptualization, methodology, laboratory experiments supervising, modeling and paper writing: A.V.S.; laboratory experiments and statistical data processing: N.B.S. and postgraduate E.A.B. All authors have read and agreed to the published version of the manuscript.

**Funding:** The theoretical studies, methodological developments, scientific equipment and main experiments were supported by the Russian Foundation for Basic Research (grant no. 19-29-05006). Nadezhda B. Sadovnikova performed laboratory analysis of hydrogels swelling in sodium chloride solutions under the state project of Moscow State University CITIS no. 121040800146-3.

**Conflicts of Interest:** The authors declare no conflict of interest. The funders had no role in the design of the study; in the collection, analyses, and interpretation of data; in the writing of the manuscript; or in the decision to publish the results.

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
