# Peer review of "Hygroscopicity of Gel-Forming Composite Materials: Thermodynamic Assessment and Technological Significance"

_jcs, doi:10.3390/jcs6090269_

Round 1

Reviewer 1 Report

In the manuscript by Smagin et al., the hygroscopicity of composite gel-forming soil conditioners was studied. Based on the thermodynamic model, the water activity and temperature influence of several different composite gel-forming materials were investigated. The obtained results provided some insight into the design of composite gel-forming with controlled hygroscopicity for different applications. I recommend accepting the manuscript after addressing the following comments.

1.      The hygroscopicity and swelling equilibrium are influenced by both material types (such as chemical composition and functional groups) and polymer network architectures (such as crosslinking density). The latter factor was not considered in this work. However, relevant discussion on this is suggested.

2.      The current introduction is too general. More detailed progress on the topic can be added.

3.      Does “80-130% (per mass)” mean “80-130 wt%”?Q

Author Response

Dear Reviewer 1!
Thank you for your positive assessment of the manuscript and comments for its improvement. We tried to take them into account in the new version. Below we answer your questions and comments in the order of the review.

  1. The hygroscopicity and swelling equilibrium are influenced by both material types (such as chemical composition and functional groups) and polymer network architectures (such as crosslinking density). The latter factor was not considered in this work. However, relevant discussion on this is suggested.

Ans: You are absolutely right, and the crosslinking density will obviously be one of the main factors controlling hygroscopicity. Unfortunately, we do not yet have experimental data concerning the effect of this factor on the hygroscopicity of hydrogels, so it cannot be considered in this paper.

  1. The current introduction is too general. More detailed progress on the topic can be added.

Ans: We took into account your comment and tried to detail a brief overview in the introduction.

  1. Does “80-130% (per mass)” mean “80-130 wt%”?Q

Ans: Yes, you're right. We have made corrections in the text.

Reviewer 2 Report

The paper constitutes an interesting and valuable work. The only revisions which are strongly suggested are as follows:

1) Section Introduction: Authors mentioned that: "Innovative composite materials can be used both in water purification technologies and for its conservation in the soil by reducing unproductive losses due to evaporation and infiltration [2-11]." This sentence is supported by 10 references. This is not proper. Authors should significantly extend the discussion and avoid basing on such big group of references.

2) Section Conclusions should be shortened. Additionally, this section needs to be supplemented with some quantified data.

3) Section References should be corrected and prepared according to the requirements of the Journal, e.g. the whole journal names should be replaced by their abbreviations.

Author Response

Dear Reviewer 2!

Thank you very much for positive assessment of the manuscript and comments for its improvement. We tried to take them into account in the new version. Below we answer your questions and comments in order of review:

  • Section Introduction: Authors mentioned that: "Innovative composite materials can be used both in water purification technologies and for its conservation in the soil by reducing unproductive losses due to evaporation and infiltration [2-11]." This sentence is supported by 10 references. This is not proper. Authors should significantly extend the discussion and avoid basing on such big group of references.

Ans: We agree with you and have tried to expand the overview part of the introduction by specifying this basic provision

  • Section Conclusions should be shortened. Additionally, this section needs to be supplemented with some quantified data.

Ans: We have tried to reduce and supplement the conclusions with some quantitative data, according to your comment

  • Section References should be corrected and prepared according to the requirements of the Journal, e.g. the whole journal names should be replaced by their abbreviations.

Ans: We checked the reference list and took into account your remark, replacing the full whole journal names should be replaced by their abbreviations.

From the team of authors, sincerely yours, Prof. Andrey Smagin

Round 2

Reviewer 2 Report

The paper has been corrected. Authors considered all recommendations. The manuscript in its current version may be accepted for publication.

Author Response

Dear Reviewers! On behalf of the team of authors, I sincerely thank you for the work on reviewing the manuscript and its positive assessment. With best wishes, prof. Andrey Smagin
